# Training independent subnetworks for robust prediction

**Marton Havasi**[*]
Department of Engineering
University of Cambridge
mh740@cam.ac.uk

**Rodolphe Jenatton**
Google Research
rjenatton@google.com

**Stanislav Fort**
Stanford University
sfort1@stanford.edu

**Jeremiah Zhe Liu**
Google Research &
Harvard University
jereliu@google.com

**Jasper Snoek**
Google Research
jsnoek@google.com

**Balaji Lakshminarayanan**
Google Research
balajiln@google.com

**Andrew M. Dai**
Google Research
adai@google.com

**Dustin Tran**
Google Research
trandustin@google.com

## Abstract

Recent approaches to efficiently ensemble neural networks have shown that strong robustness and uncertainty performance can be achieved with a negligible gain in parameters over the original network. However, these methods still require multiple forward passes for prediction, leading to a significant computational cost. In this work, we show a surprising result: the benefits of using multiple predictions can be achieved 'for free' under a single model's forward pass. In particular, we show that, using a multi-input multi-output (MIMO) configuration, one can utilize a single model's capacity to train multiple subnetworks that independently learn the task at hand. By ensembling the predictions made by the subnetworks, we improve model robustness without increasing compute. We observe a significant improvement in negative log-likelihood, accuracy, and calibration error on CIFAR10, CIFAR100, ImageNet, and their out-of-distribution variants compared to previous methods.

## 1 Introduction

Uncertainty estimation and out-of-distribution robustness are critical problems in machine learning. In medical applications, a confident misprediction may be a misdiagnosis that is not referred to a physician as during decision-making with a "human-in-the-loop." This can have disastrous consequences, and the problem is particularly challenging as patient data deviates significantly from the training set such as in demographics, disease types, epidemics, and hospital locations (Dusenberry et al., 2020b; Filos et al., 2019).

Using a distribution over neural networks is a popular solution stemming from classic Bayesian and ensemble learning literature (Hansen & Salamon, 1990; Neal, 1996), and recent advances such as BatchEnsemble and extensions thereof achieve strong uncertainty and robustness performance (Wen et al., 2020; Dusenberry et al., 2020a; Wenzel et al., 2020). These methods demonstrate that significant gains can be had with negligible additional parameters compared to the original model. However, these methods still require multiple (typically, 4-10) forward passes for prediction, leading to a significant runtime cost. In this work, we show a surprising result: the benefits of using multiple predictions can be achieved "for free" under a single model's forward pass.

The insight we build on comes from sparsity. Neural networks are heavily overparameterized models. The lottery ticket hypothesis (Frankle & Carbin, 2018) and other works on model pruning (Molchanov et al., 2016; Zhu & Gupta, 2017) show that one can prune away 70-80% of the connections in a

---

[*]Work done as a Google Research intern.

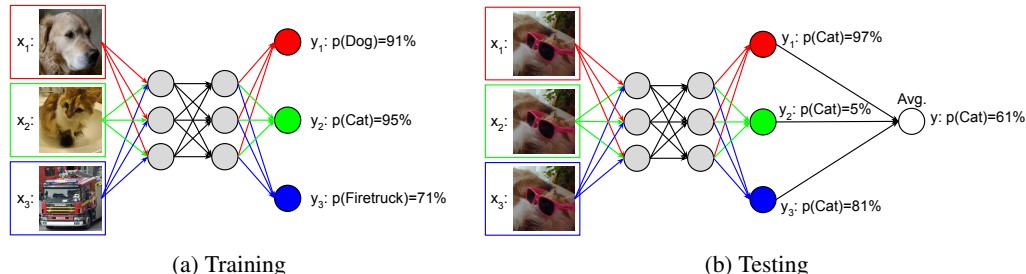

Figure 1: In the multi-input multi-output (MIMO) configuration, the network takes $M = 3$ inputs and gives $M$ outputs. The hidden layers remain unchanged. The black connections are shared by all subnetworks, while the colored connections are for individual subnetworks. (a) During training, the inputs are independently sampled from the training set and the outputs are trained to classify their corresponding inputs. (b) During testing, the same input is repeated $M$ times and the outputs are averaged in an ensemble to obtain the final prediction.

neural network without adversely affecting performance. The remaining sparse subnetwork, called the winning ticket, retains its predictive accuracy. This suggests that a neural network has sufficient capacity to fit 3-4 independent subnetworks simultaneously. We show that, using a multi-input multi-output (MIMO) configuration, *we can concurrently train multiple independent subnetworks within one network*. These subnetworks co-habit the network without explicit separation. The advantage of doing this is that at test time, we can evaluate all of the subnetworks at the same time, leveraging the benefits of ensembles in a *single forward pass*.

Our proposed MIMO configuration only requires two changes to a neural network architecture. First, replace the input layer: instead of taking a single datapoint as input, take $M$ datapoints as inputs, where $M$ is the desired number of ensemble members. Second, replace the output layer: instead of a single head, use $M$ heads that make $M$ predictions based on the last hidden layer. During training, the inputs are sampled independently from the training set and each of the $M$ heads is trained to predict its matching input (Figure 1a). Since, the features derived from the other inputs are not useful for predicting the matching input, the heads learn to ignore the other inputs and make their predictions independently. At test time, the same input is repeated $M$ times. That is, the heads make $M$ independent predictions on the same input, forming an ensemble for a single robust prediction that can be computed in a single forward pass (Figure 1b).

The core component of an ensemble's robustness such as in Deep Ensembles is the diversity of its ensemble members (Fort et al., 2019). While it is possible that a single network makes a confident misprediction, it is less likely that multiple independently trained networks make the same mistake. Our model operates on the same principle. By realizing multiple independent winning lottery tickets, we are reducing the impact of one of them making a confident misprediction. For this method to be effective, it is essential that the subnetworks make independent predictions. We empirically show that the subnetworks use disjoint parts of the network and that the functions they represent have the same diversity as the diversity between independently trained neural networks.

**Summary of contributions.**

1. We propose a multi-input multi-output (MIMO) configuration to network architectures, enabling multiple independent predictions in a single forward pass "for free." Ensembling these predictions significantly improves uncertainty estimation and robustness with minor changes to the number of parameters and compute cost.

2. We analyze the diversity of the individual members and show that they are as diverse as independently trained neural networks.

3. We demonstrate that when adjusting for wall-clock time, MIMO networks achieve new state-of-the-art on CIFAR10, CIFAR100, ImageNet, and their out-of-distribution variants.

## 2 MULTI-INPUT MULTI-OUTPUT NETWORKS

The MIMO model is applicable in a supervised classification or regression setting. Denote the set of training examples $\mathbb{X} = \{(\boldsymbol{x}^{(n)}, \boldsymbol{y}^{(n)})\}_{n=1}^{N}$ where $\boldsymbol{x}^{(n)}$ is the $n^{th}$ datapoint with the corresponding label $\boldsymbol{y}^{(n)}$ and $N$ is the size of the training set. In the usual setting, for an input $\boldsymbol{x}$, the output

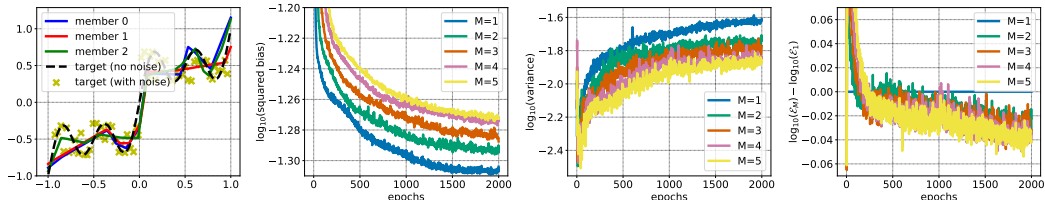

Figure 2: Illustration of MIMO applied to a synthetic regression problem. (left) Example of MIMO learning $M = 3$ diverse predictors. As $M$ increases, predicting with MIMO comes with a higher bias but a smaller variance (two middle panels respectively). Despite the slight increase in bias, the decrease in variance translates into an improved generalization performance (right).

of the neural network $\hat{\mathbf{y}}$ is a probability distribution $p_\theta(\hat{\mathbf{y}}|\boldsymbol{x})$,[1] which captures the uncertainty in the predictions of the network. The network parameters $\theta$ are trained using stochastic gradient descent (SGD) to minimize the loss $L(\theta)$ on the training set, where the loss usually includes the negative log-likelihood and a regularization term $R$ (such as the L2 regularization): $L(\theta) = \mathbb{E}_{(\mathbf{x},\mathbf{y})\in\mathbb{X}}[-\log p_\theta(\mathbf{y}|\boldsymbol{x})] + R(\theta)$.

In the MIMO configuration, the network takes $M$ inputs and returns $M$ outputs (Figure 1), where each output is a prediction for the corresponding input. This requires two small changes to the architecture. At the input layer, the $M$ inputs $\{\mathbf{x}_1, \ldots, \mathbf{x}_M\}$ are concatenated before the first hidden layer is applied and at the output layer, the network gives $M$ predictive distributions $\{p_\theta(\mathbf{y}_1|\boldsymbol{x}_1, \ldots, \boldsymbol{x}_M), \ldots, p_\theta(\mathbf{y}_M|\boldsymbol{x}_1, \ldots, \boldsymbol{x}_M)\}$ correspondingly. Having $M$ input and $M$ outputs require additional model parameters. The additional weights used in the MIMO configuration account for just a 0.03 % increase in the total number of parameters and 0.01 % increase in floating-point operations (FLOPs).[2]

The network is trained similarly to a traditional neural network, with a few key modifications to account for the $M$ inputs and $M$ outputs (Figure 1a). During training, the inputs $\mathbf{x}_1, \ldots, \mathbf{x}_M$ are sampled independently from the training set. The loss is the sum of the negative log-likelihoods of the predictions and the regularization term:

$$L_M(\theta) = \mathop{\mathbb{E}}_{\substack{(\boldsymbol{x}_1, \boldsymbol{y}_1)\in\mathbb{X} \\ \cdots \\ (\boldsymbol{x}_M, \boldsymbol{y}_M)\in\mathbb{X}}} \left[\sum_{m=1}^{M} -\log p_\theta(\boldsymbol{y}_m|\boldsymbol{x}_1, \ldots, \boldsymbol{x}_M)\right] + R(\theta),$$

which is optimized using stochastic gradient descent. Note that the sum of the log-likelihoods is equal to the log-likelihood of the joint distribution $\sum_{m=1}^{M} \log p_\theta(\boldsymbol{y}_m|\boldsymbol{x}_1, \ldots, \boldsymbol{x}_M) = \log p_\theta(\boldsymbol{y}_1, \ldots, \boldsymbol{y}_M|\boldsymbol{x}_1, \ldots, \boldsymbol{x}_M)$ since the input-output pairs are independent. Hence a second interpretation of MIMO is that it is simply training a traditional neural network over $M$-tuples of independently sampled datapoints.

At evaluation time, the network is used to make a prediction on a previously unseen input $\boldsymbol{x}'$. The input $\boldsymbol{x}'$ is tiled $M$ times, so $\boldsymbol{x}_1 = \ldots = \boldsymbol{x}_M = \boldsymbol{x}'$ (Figure 1b). Since all of the inputs are $\boldsymbol{x}'$, each of the outputs independently approximate the predictive distribution $p_\theta(\mathbf{y}_m|\boldsymbol{x}', \ldots, \boldsymbol{x}') \approx p(\mathbf{y}'|\boldsymbol{x}')$ (for $m = 1 \ldots M$). As an ensemble, averaging these predictions improves the predictive performance, leading to the combined output $p_\theta(\mathbf{y}'|\boldsymbol{x}') = \frac{1}{M}\sum_{m=1}^{M} p_\theta(\mathbf{y}_m|\boldsymbol{x}', \ldots, \boldsymbol{x}')$.

Unlike Bayesian methods requiring multiple weight samples, or even parameter-efficient methods like BatchEnsemble, MIMO's advantage is that all of the ensemble members can be calculated in a single forward pass. As a result, MIMO's wall-clock time is almost equivalent to a standard neural network.

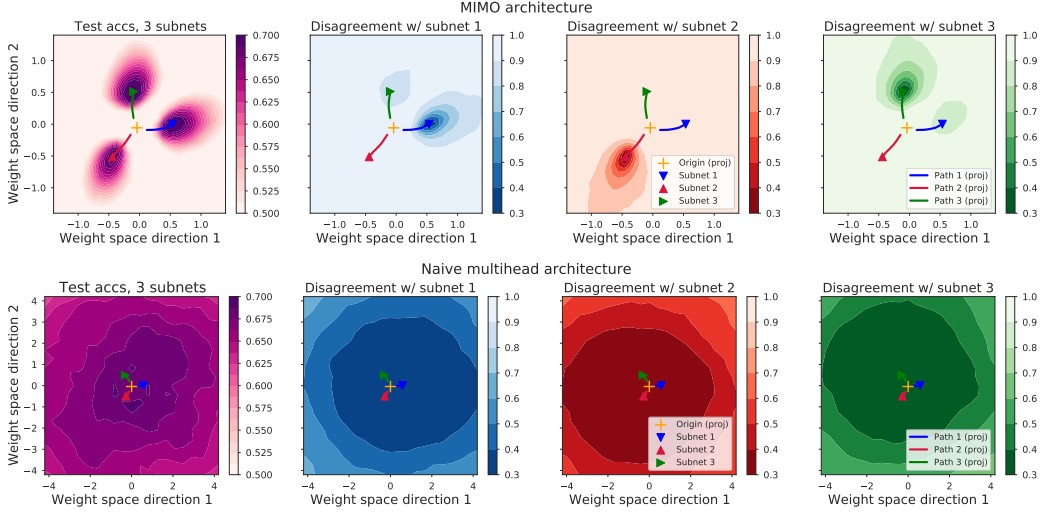

Figure 3: Accuracy landscape and function space landscape comparison of individual subnetworks for MIMO (top row) and the naive multiheaded architecture (bottom row). (left): The test accuracy in the weight space section containing $M = 3$ trained subnetworks and the origin. For the MIMO architecture, the individual subnetworks converge to three distinct low-loss basins, while naive multihead leads to the same mode. (middle-left to right): The blue, red and green panels show the disagreement between the three trained subnetworks for the same section of the weight space. For the MIMO architecture, the subnetworks often disagree, while for the naive multihead architecture they are all essentially equivalent.

## 2.1 ILLUSTRATION OF MIMO ON A SYNTHETIC REGRESSION EXAMPLE

Before applying MIMO to large-scale vision models, we first illustrate its behavior on a simple one-dimensional regression problem. We consider the noisy function from Blundell et al. (2015) (see Figure 2, left), with a training and test set of $N = 64$ and 3000 observations respectively. We train a multilayer perceptron with two hidden-layers, composed of (32, 128) units and ReLU activations. [3]

For different ensemble sizes $M \in \{1, \ldots, 5\}$, we evaluate the resulting models in terms of expected mean squared error $\mathcal{E}_M$. If we denote by $\hat{f}_M$ the regressor with $M$ ensemble members learned over $\mathbb{X}$, we recall that $\mathcal{E}_M = \mathbb{E}_{(\boldsymbol{x}', y') \in \mathbb{X}_{\text{test}}}[\mathbb{E}_{\mathbb{X}}[(\hat{f}_M(\boldsymbol{x}', \ldots, \boldsymbol{x}') - y')^2]]$, where $\mathbb{E}_{\mathbb{X}}[\cdot]$ denotes the expectation over training sets of size $N$. We make two main observations.

First, in the example of $M = 3$ in Figure 2 (left), we can see that MIMO can learn a diverse set of predictors. Second, the diverse predictors obtained by MIMO translate into improved performance, as seen in Figure 2 (right). Moreover, in the regression setting, we can decompose $\mathcal{E}_M$ into its (squared) bias and variance components (Sec. 2.5 in Hastie et al. (2009)). More formally, we have

$$\mathcal{E}_M = \underbrace{\mathbb{E}_{(\boldsymbol{x}', y') \in \mathbb{X}_{\text{test}}}\left[(\bar{f}_M(\boldsymbol{x}', \ldots, \boldsymbol{x}') - y')^2\right]}_{\text{(squared) bias}} + \underbrace{\mathbb{E}_{(\boldsymbol{x}', y') \in \mathbb{X}_{\text{test}}}\left[\mathbb{E}_{\mathbb{X}}\left[(\bar{f}_M(\boldsymbol{x}', \ldots, \boldsymbol{x}') - \hat{f}_M(\boldsymbol{x}', \ldots, \boldsymbol{x}'))^2\right]\right]}_{\text{variance}},$$

where $\bar{f}_M = \mathbb{E}_{\mathbb{X}}[\hat{f}_M]$. The bias-variance decomposition nicely captures the strength of MIMO. While learning a neural network over $M$-tuples induces a slight bias compared to a standard model with $M = 1$, i.e. the individual members perform slightly worse (Figure 2, middle-left), this is compensated by the diverse predictions of the ensemble members that lead to lower variance (Figure 2, middle-right). MIMO yields an improvement when the model has sufficient capacity to fit $M > 1$ diverse, well-performing ensemble members.

---

[1] We use bold font to denote random variables and italic to denote their instantiations, for example, random variable $\mathbf{z}$ and its instantiation $\boldsymbol{z}$.

[2] A standard ResNet28-10 has 36.479M parameters and it takes 10.559G FLOPs to evaluate, while in the MIMO configuration, it has 36.492M parameters and takes 10.561G FLOPs to evaluate.

[3] We provide an interactive notebook that reproduces these experiments.

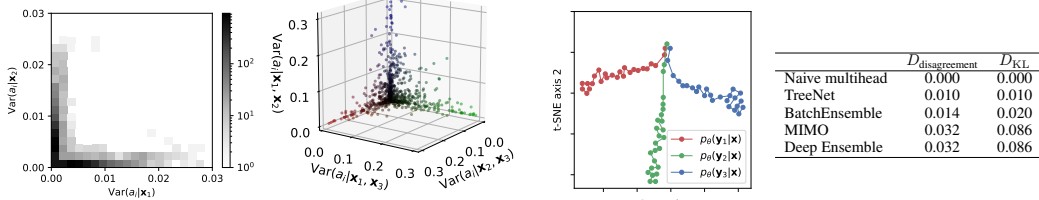

Figure 4: Analyzing the subnetworks on the CIFAR10 dataset. (left): Histogram of the conditional variances of the pre-activations w.r.t. each input ($M = 2$, ResNet28-10). (middle-left): Scatter plot of the conditional variances of the pre-activations w.r.t. each input. Almost all the pre-activations only have variance with respect to one of the inputs: the subnetwork they that are part of ($M = 3$, ResNet28-10). (middle-right): Training trajectories of the subnetworks. The subnetworks converge to different local optima ($M = 3$, SmallCNN). (right): Diversity of the members ($\mathcal{D}_D$) in different efficient ensemble models (ResNet 28-10).

## 3 UNDERSTANDING THE SUBNETWORKS

The mapping of each input-output pair in a MIMO configuration is referred to as a subnetwork. In this section, we show that the subnetworks converge to distinct local optima and they functionally behave as independently trained neural networks.

### 3.1 LOSS-LANDSCAPE ANALYSIS

Using multiple inputs is the key to training diverse subnetworks. The subnetworks learn independently, since features derived from each input are only useful for the corresponding output. In contrast, in a *naive multiheaded* architecture, where the input is shared, but the model has separate $M$ outputs, the outputs rely on the same features for prediction, which leads to very low diversity.

To showcase this, we replicate a study from Fort et al. (2019). We look at the SmallCNN model (3 convolutional layers with 16, 32, 32 filters respectively) trained on CIFAR-10, and linearly interpolate between the three subnetworks in weight space. In the case of MIMO, we interpolate the input and output layers, since the body of the network is shared, and for the naive multiheaded model, we only interpolate the output layers, since the input and the body of the network is shared. Analogously to Deep Ensembles, the subnetworks trained using MIMO converge to disconnected modes in weight space due to differences in initialization, while in the case of the naive multiheaded model, the subnetworks end up in the same mode (Figure 3, left). Figure 3 (right) shows the disagreement i.e. the probability that the subnetworks disagree on the predicted class. MIMO's disconnected modes yield diverse predictions, while the predictions of the naive multiheaded model are highly correlated.

### 3.2 FUNCTION SPACE ANALYSIS

We visualize the training trajectories of the subnetworks in MIMO in function space (similarly to Fort et al. (2019)). As we train the SmallCNN architecture ($M = 3$), we periodically save the predictions on the test set. Once training is finished, we plot the t-SNE projection (Maaten & Hinton, 2008) of the predictions. We observe that the trajectories converge to distinct local optima (Figure 4).

For a quantitative measurement of diversity in large scale networks, we look at the average pairwise similarity of the subnetwork's predictions at test time, and compare against other efficient ensemble methods. The average pairwise similarity is

$$\mathcal{D}_D = \mathbb{E}\left[D\left(p_\theta(\mathbf{y}_1|\mathbf{x}, \mathbf{x}\dots\mathbf{x}), p_\theta(\mathbf{y}_2|\mathbf{x}, \mathbf{x}\dots\dots\mathbf{x})\right)\right],$$

where $D$ is a distance metric between predictive distributions and $(\mathbf{x}, \mathbf{y}) \in \mathbb{X}$. We consider two distance metrics. Disagreement: whether the predicted classes agree, $D_{\text{disagreement}}(P_1, P_2) = \mathbb{I}(\arg\max_{\hat{y}} P_1(\hat{y}) = \arg\max_{\hat{y}} P_2(\hat{y}))$ and Kullback–Leibler divergence: $D_{\text{KL}}(P_1, P_2) = \mathbb{E}_{P_1}[\log P_1(y) - \log P_2(y)]$. When the ensemble members give the same prediction at all test points, both their disagreement and KL divergence are 0.

The first efficient ensemble approach we compare against is the aforementioned *naive multiheaded* model, where the input and the body of the neural network is shared by the ensemble members, but each member has its own output layer. Next, *TreeNet* (Lee et al., 2015), where the input and the first two residual groups are shared, but the final residual group and the output layer are trained separately for each member. Finally, *BatchEnsemble* (Wen et al., 2020), where the members share network parameters up to a rank-1 perturbation, which changes information flow through the full network.

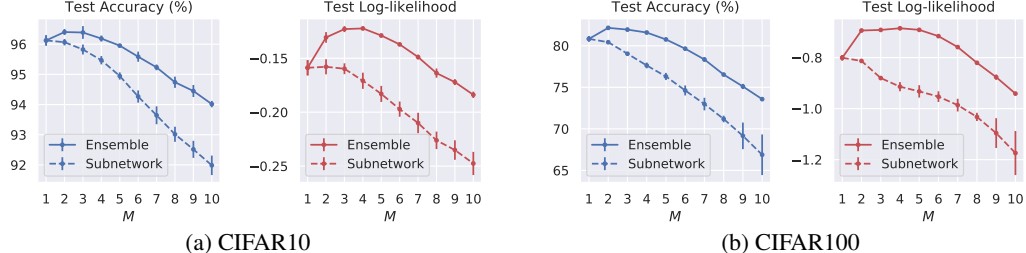

Figure 5: The performance of the subnetworks and the ensemble of the subnetworks as the number of subnetworks ($M$) varies. $M = 1$ is equivalent to a standard neural network (ResNet-28-10).

We find that the naive multiheaded model fails to induce diversity: the predictions of its subnetworks are nearly identical on all test points as shown in Figure 4 (right). TreeNet and BatchEnsemble have more diversity, although there is still significant correlation in the predictions. MIMO has better diversity than prior efficient ensemble approaches and it matches the diversity of independently trained neural networks.

These results allow us to pinpoint the source of robustness: The robustness of MIMO comes from ensembling the diverse predictions made by the subnetworks, thus MIMO faithfully replicates the behaviour of a Deep Ensemble within one network.

### 3.3 SEPARATION OF THE SUBNETWORKS

To show that the subnetworks utilize separate parts of the network, we look at the activations and measure how they react to changes in each of the $M$ inputs. Namely, we calculate the conditional variance of each pre-activation in the network with respect to each individual input. For input $\mathbf{x}_1$:

$$\mathrm{Var}(a_i|\mathbf{x}_2) = \mathop{\mathbb{E}}_{\mathbf{x}_2}[\mathop{\mathrm{Var}}_{\boldsymbol{x}_1}(a_i(\mathbf{x}_1, \boldsymbol{x}_2)] \quad (M=2), \text{ and } \mathrm{Var}(a_i|\mathbf{x}_2, \mathbf{x}_3) = \mathop{\mathbb{E}}_{\mathbf{x}_2, \mathbf{x}_3}[\mathop{\mathrm{Var}}_{\boldsymbol{x}_1}(a_i(\mathbf{x}_1, \boldsymbol{x}_2, \boldsymbol{x}_3)] \quad (M=3)$$

where $a_i$ is the value of $i$-th pre-activation in the function of the $M$ inputs. For reference, there are 8190 pre-activations in a ResNet28-10. We can estimate the conditional variance by fixing $\mathbf{x}_1$ and calculating the variance of $a_i$ w.r.t. $\mathbf{x}_2 \in \mathbb{X}$ (when $M = 2$, $\mathbf{x}_2, \mathbf{x}_3 \in \mathbb{X}$ when $M = 3$) and finally averaging over the possible values $\mathbf{x}_1 \in \mathbb{X}$. The conditional variance is analogously defined for $\mathbf{x}_2, \ldots, \mathbf{x}_M$. If the conditional variance of an activation is non-zero w.r.t. an input, that means that the activation changes as the input changes and therefore we consider it part of the subnetwork corresponding to the input. If the subnetworks are independent within the network, we expect that the conditional variance of each activation is non-zero w.r.t. one of the inputs, the subnetwork to which it belongs, and close-to zero w.r.t. all the other inputs.

When we plot the conditional variances, this is exactly what we see. In Figure 4 (left), we see the histogram of the pre-activations in the network. Each point has two corresponding values: the conditional variance w.r.t. the two inputs. As we can see, all activations have non-zero conditional variance w.r.t. one of the inputs and close-to zero w.r.t. the other. Figure 4 (middle-left) shows the scatterplot of the activations for $M = 3$. We see that, similarly to $M = 2$, almost all activations have non-zero conditional variance w.r.t. one of the inputs and close-to zero conditional variance w.r.t. the others. Since almost all activations are part of exactly one of the subnetworks, which we can identify by calculating the conditional variances, we conclude that the subnetworks separate within the network. This implies an extension to Frankle & Carbin (2018): the subnetworks realize separate winning lottery tickets within a single network instance.

### 3.4 THE OPTIMAL NUMBER OF SUBNETWORKS

A natural question that arises is the ideal number of subnetworks $M$ to fit in a network. Too few subnetworks do not fully leverage the benefits of ensembling, while having too many quickly reaches the network capacity, hurting their individual performances. Ideally, we are looking to fit as many subnetworks as possible without significantly impacting their individual performances.

Figure 5 shows the performance of both the individual subnetworks and the ensemble as $M$ varies. $M = 1$ is equivalent to a traditionally trained network i.e. the performance of the subnetwork matches the performance of the ensemble since there is only one subnetwork. As $M$ grows, we can see that the performance of the subnetworks slowly declines as they utilize more and more of the network

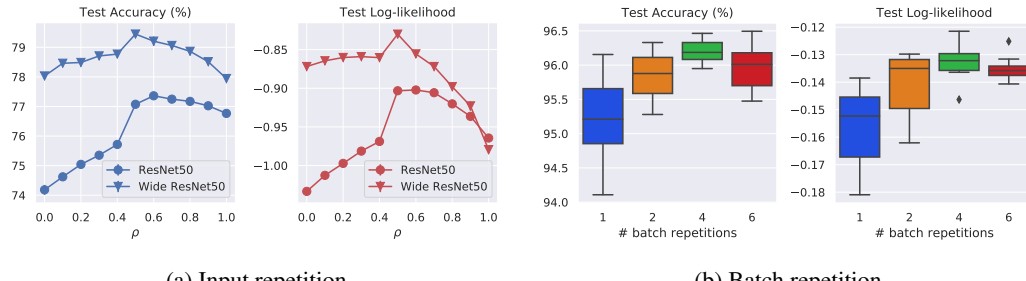

(a) Input repetition          (b) Batch repetition

Figure 6: (a) Performance of MIMO ($M = 2$) as a function of $\rho$ on ImageNet. At $\rho = 0$, the subnetworks are independent and they are limited by the network capacity. With $\rho > 0$, the subnetworks are able to share features and better utilize the network capacity. Wide ResNet has $2\times$ more filters. (b) Repeating examples in the same batch improves convergence and yields a slight boost in performance.

capacity. The performance of the ensemble, however, peaks between $M = 2$ and $M = 4$, where the benefits of ensembling outweigh the slight decrease in performance of the individual subnetworks. Interestingly, the accuracy peaks earlier than the log-likelihood, which suggests that ensembling is more beneficial for the latter.

In Appendix C, we further illustrate how MIMO exploits the capacity of the network. In particular, we study the performance of MIMO when the regularization increases (both in terms of L1 and L2 regularization), i.e. when the capacity of the network is increasingly constrained. In agreement with our hypothesis that MIMO better utilizes capacity, we observe that its performance degrades more quickly as the regularization intensifies. Moreover, the larger $M$, the stronger the effect. Interestingly, for the L1 regularization, we can relate the performance of MIMO with the *sparsity* of the network, strengthening the connection to Frankle & Carbin (2018).

### 3.5  INPUT AND BATCH REPETITION

MIMO works well by simply adding the multi-input and multi-output configuration to an existing baseline, and varying only one additional hyperparameter (Section 3.4's number of subnetworks $M$). We found two additional hyperparameters can further improve performance, and they can be important when the network capacity is limited.

**Input repetition** Selecting independent examples for the multi-input configuration during training forces the subnetworks not to share any features. This is beneficial when the network has sufficient capacity, but when the network has limited excess capacity, we found that relaxing independence is beneficial. For example, ResNet50 on ImageNet (He et al., 2016) does not have sufficient capacity to support two independent subnetworks ($M = 2$) in MIMO configuration.

Our proposed solution is to relax independence between the inputs. Instead of independently sampling $\mathbf{x}_1$ and $\mathbf{x}_2$ from the training set during training, they share the same value with probability $\rho$. That is, $\mathbf{x}_1$ is sampled from the training set and $\mathbf{x}_2$ is set to be equal to $\mathbf{x}_1$ with probability $\rho$ or sampled from the training set with probability $1 - \rho$. Note that this does not affect the marginal distributions of $\mathbf{x}_1$ and $\mathbf{x}_2$, it merely introduces a correlation in their joint distribution.

Figure 6a shows the performance as $\rho$ varies. At $\rho = 0$, the subnetworks are independent but their performance is limited by the network capacity. As $\rho$ grows, the subnetworks share increasingly more features, which improves their performance. However, as $\rho$ approaches 1, the subnetworks become highly correlated and the benefit of ensembling is lost. Unlike ResNet50, Wide ResNet50 has more capacity and benefits less from input repetition (roughly 78-79% vs 74-77%).

**Batch repetition** For stochastic models, most notably MC dropout and variational Bayesian neural nets, drawing multiple approximate posterior samples for each example during training can improve performance as it reduces gradient noise w.r.t. the network's model uncertainty, e.g., Dusenberry et al. (2020a). We achieve a similar effect by repeating examples in the minibatch: this forms a new minibatch size of, e.g., $512 \cdot 5$ (batch size and number of batch repetitions respectively). Like the choice of batch size which determines the number of unique examples in the SGD step (Shallue et al., 2018), varying the number of repetitions has an implicit regularization effect. Figure 6b shows performance over the number of batch repetitions, where each batch repetition setting indicates a box

plot over a sweep of 12 settings of batch size, learning rate, and ensemble size. Higher repetitions typically yield a slight boost.

## 4    BENCHMARKS

We described and analyzed MIMO. In this section, we compare MIMO on benchmarks building on Uncertainty Baselines.[4] This framework allows us to benchmark the performance and to compare against high-quality, well-optimized implementations of baseline methods (see framework for further baselines than ones highlighted here). We looked at three model/dataset combinations: ResNet28-10/CIFAR10, ResNet28-10/CIFAR100, and ResNet50/ImageNet. MIMO's code is open-sourced. [5]

**Baselines** Our baselines include the reference implementations of a deterministic deep neural network (trained with SGD), MC-Dropout, BatchEnsemble, and ensemble models, as well as two related models, Naive multihead and TreeNet. Thin networks use half the number of convolutional filters while wide models use double. See Appendix B for the details on the hyperparameters.

**Metrics** To measure robustness, we look at accuracy, negative log-likelihood (NLL), and expected calibration error (ECE) on the IID test set as well as a corrupted test set where the test images are perturbed (e.g. added blur, compression artifacts, frost effects) (Hendrycks & Dietterich, 2019). Appendix D includes ImageNet results for 5 additional out-of-distribution datasets. To measure computational cost, we look at how long it takes to evaluate the model on a TPUv2 core, measured in ms per example.

| Name | Accuracy (↑) | NLL (↓) | ECE (↓) | cAcc (↑) | cNLL (↓) | cECE (↓) | Prediction time (↓) | # Forward passes (↓) |
|---|---|---|---|---|---|---|---|---|
| Deterministic | 96 | 0.159 | 0.023 | 76.1 | 1.050 | 0.153 | 0.632 | 1 |
| Dropout | 95.9 | 0.160 | 0.024 | 68.8 | 1.270 | 0.166 | 0.656 | 1 |
| Naive mutlihead ($M = 3$) | 95.9 | 0.161 | 0.022 | **76.6** | 0.969 | 0.144 | 0.636 | 1 |
| MIMO ($M = 3$) (This work) | **96.4** | **0.123** | **0.010** | 76.6 | **0.927** | **0.112** | 0.639 | 1 |
| TreeNet ($M = 3$) | 95.9 | 0.158 | 0.018 | 75.5 | 0.969 | 0.137 | 0.961 | 1.5 |
| BatchEnsemble ($M = 4$) | 96.2 | 0.143 | 0.021 | 77.5 | 1.020 | 0.129 | 2.552 | 4 |
| Thin Ensemble ($M = 4$) | 96.3 | 0.115 | 0.008 | 77.2 | 0.840 | 0.089 | 0.823 | 4 |
| Ensemble ($M = 4$) | 96.6 | 0.114 | 0.010 | 77.9 | 0.810 | 0.087 | 2.536 | 4 |

Table 1: ResNet28-10/CIFAR10: The best single forward pass results are highlighted in bold.

| Name | Accuracy (↑) | NLL (↓) | ECE (↓) | cAcc (↑) | cNLL (↓) | cECE (↓) | Prediction time (↓) | # Forward passes (↓) |
|---|---|---|---|---|---|---|---|---|
| Deterministic | 79.8 | 0.875 | 0.086 | 51.4 | 2.700 | 0.239 | 0.632 | 1 |
| Monte Carlo Dropout | 79.6 | 0.830 | 0.050 | 42.6 | 2.900 | 0.202 | 0.656 | 1 |
| Naive mutlihead ($M = 3$) | 79.5 | 0.834 | 0.048 | 52.1 | 2.339 | 0.156 | 0.636 | 1 |
| MIMO ($M = 3$) (This work) | **82.0** | **0.690** | **0.022** | **53.7** | **2.284** | **0.129** | 0.639 | 1 |
| TreeNet ($M = 3$) | 80.8 | 0.777 | 0.047 | 53.5 | 2.295 | 0.176 | 0.961 | 1.5 |
| BatchEnsemble ($M = 4$) | 81.5 | 0.740 | 0.056 | 54.1 | 2.490 | 0.191 | 2.552 | 4 |
| Thin Ensemble ($M = 4$) | 81.5 | 0.694 | 0.017 | 53.7 | 2.190 | 0.111 | 0.823 | 4 |
| Ensemble ($M = 4$) | 82.7 | 0.666 | 0.021 | 54.1 | 2.270 | 0.138 | 2.536 | 4 |

Table 2: ResNet28-10/CIFAR100: The best single forward pass results are highlighted in bold.

| Name | Accuracy (↑) | NLL (↓) | ECE (↓) | cAcc (↑) | cNLL (↓) | cECE (↓) | Prediction time (↓) | # Forward passes (↓) |
|---|---|---|---|---|---|---|---|---|
| Deterministic | 76.100 | 0.943 | 0.039 | 40.500 | 3.200 | **0.105** | 0.640 | 1 |
| Naive mutlihead ($M = 3$) | 76.611 | 0.929 | 0.043 | 40.616 | 3.250 | 0.122 | 0.638 | 1 |
| MIMO ($M = 2$) ($\rho = 0.6$) (This work) | **77.500** | **0.887** | **0.037** | **43.300** | **3.030** | 0.106 | 0.635 | 1 |
| TreeNet ($M = 2$) | 78.139 | 0.852 | 0.017 | 42.420 | 3.052 | 0.073 | 0.848 | 1.5 |
| BatchEnsemble ($M = 4$) | 76.700 | 0.944 | 0.049 | 41.800 | 3.180 | 0.110 | 2.592 | 4 |
| Ensemble ($M = 4$) | 77.500 | 0.877 | 0.031 | 42.100 | 2.990 | 0.051 | 2.624 | 4 |
| Wide Deterministic | 77.885 | 0.938 | 0.072 | 45.000 | 3.100 | 0.150 | 1.674 | 1 |
| Wide MIMO ($M = 2$) ($\rho = 0.6$) (This work) | 79.300 | 0.843 | 0.061 | 45.791 | 3.048 | 0.147 | 1.706 | 1 |

Table 3: ResNet50/ImageNet: The best single forward pass results are highlighted in bold.

The metrics show that MIMO significantly outperforms other single forward pass methods on all three benchmarks. It approaches the robustness of a Deep Ensemble, without increasing the computational costs.

---

[4]https://github.com/google/uncertainty-baselines
[5]https://github.com/google/edward2/tree/master/experimental/mimo

## 5 RELATED WORK

Multi-headed networks have been previously studied by Lee et al. (2015); Osband et al. (2016); Tran et al. (2020). In this approach to efficient ensembles, the input and part of the network are shared by the members while the final few layers and the outputs are separate. The advantage of the approach is that the computational cost is reduced compared to typical ensembles, since many layers are shared, but the ensemble diversity (and resulting performance) is quite lacking. MIMO's multi-input configuration makes a significant impact as each ensemble member may take different paths throughout the full network. Further, MIMO has lower computational cost than multi-headed approaches, since all of the layers except the first and last are shared.

Related efficient ensemble approaches include BatchEnsemble, Rank-1 BNNs, and hyper batch ensembles (Wen et al., 2020; Dusenberry et al., 2020a; Wenzel et al., 2020). In these methods, most of the model parameters are shared among the members, which reduces the memory requirement of the model, but the evaluation cost still requires multiple forward passes. Interestingly, like MIMO, these methods also apply a multi-input multi-output configuration, treating an ensemble of networks as a single bigger network; however, MIMO still outperforms BatchEnsemble. We believe important insights such as Section 3.5's input independence may also improve these methods.

Finally, there are simple heuristics which also retain efficient compute such as data augmentation, temperature scaling, label smoothing, contrastive training. These methods are orthogonal to MIMO and they can provide a performance boost, without increasing the computational cost.

## 6 CONCLUSIONS

We propose MIMO, a novel approach for training multiple independent subnetworks within a network. We show that the subnetworks separate within the model and that they behave as independently trained neural networks. The key benefits of MIMO are its simplicity, since it does not require significant modifications to the network architecture and it has few hyperparameters, and also its computational efficiency, since it can be evaluated in a single forward pass. Our empirical results confirm that MIMO improves performance and robustness with minor changes to the number of parameters and the compute cost.

## ACKNOWLEDGEMENTS

Marton Havasi is funded by EPSRC. We thank Ellen Jiang for helping with writing.

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

## A PSEUDOCODE

---

**Algorithm 1** Train($\mathbb{X}$)

---

1: **for** $t = 1 \ldots N_{\text{iter}}$ **do**
2:      $(\boldsymbol{x}_{1:M}, \boldsymbol{y}_{1:M}) \sim U(\mathbb{X})$
3:      $p_\theta(\mathbf{y}_1|\boldsymbol{x}_{1:M}) \ldots p_\theta(\mathbf{y}_M|\boldsymbol{x}_{1:M}) \leftarrow \text{MIMO}(\boldsymbol{x}_{1:M})$
4:      $L_M(\theta) \leftarrow \sum_{m=1}^{M} -\log p_\theta(\boldsymbol{y}_m|\boldsymbol{x}_{1:M}) + R(\theta)$
5:      $\theta \leftarrow \theta - \epsilon \nabla L_M(\theta)$                        ▷ $\epsilon$ is the learning rate.
6: **end for**

---

**Algorithm 2** Evaluate($\boldsymbol{x}'$)

---

1: $p_\theta(\mathbf{y}_1|\mathbf{x}_{1:M} = \boldsymbol{x}') \ldots p_\theta(\mathbf{y}_M|\mathbf{x}_{1:M} = \boldsymbol{x}') \leftarrow \text{MIMO}(\mathbf{x}_{1:M} = \boldsymbol{x}')$
2: **return** $\frac{1}{M} \sum_{m=1}^{M} p_\theta(\mathbf{y}_m|\mathbf{x}_{1:M} = \boldsymbol{x}')$

---

## B HYPERPARAMETERS

For the ResNet28-10/CIFAR models, we use a batch-size of 512, a decaying learning rate of 0.1 (decay rate 0.1) and L2 regularization 2e-4. The Deterministic, Dropout and Ensemble models are trained for 200 epochs while BatchEnsemble, Naive multihead and TreeNet are trained for 250 epochs.

For the ResNet50/ImageNet models, we use a batch-size of 4096 and a decaying learning rate of 0.1 (decay rate 0.1) and L2 regularization 1e-4. The Deterministic, Dropout and Ensemble models are trained for 90 epochs, the BatchEnsemble model is trained for 135 epochs and Naive multihead and TreeNet are trained for 150 epochs.

Regarding model specific hyperparameters, Dropout uses a 10% dropout rate and a single forward pass at evaluation time. Both Ensemble and BatchEnsemble models use $M = 4$ members, since this provides most of the benefits of ensembling without significantly increasing the computational costs. The TreeNet architecture

**MIMO** For MIMO, we use the hyperparameters of the baseline implementations wherever possible. For the ResNet28-10/CIFAR models, we use a batch-size of 512 with decaying learning rate of 0.1 (decay rate 0.1), L2 regularization 3e-4, 250 training epochs, and a batch repetition of 4. For the ResNet50/ImageNet models, we use a batch-size of 4096 with decaying learning rate of 0.1 (decay rate 0.1), L2 regularization 1e-4, 150 training epochs, and batch repetition of 2. This makes the training cost of MIMO comparable to that of BatchEnsemble and Ensemble models. For the ResNet28-10/CIFAR experiments, we use $M = 3$ subnetworks because it performs well in both accuracy and log-likelihood. For ResNet50/ImageNet the model has lower capacity so we used $M = 2$ with $\rho = 0.6$.

## C MIMO BETTER EXPLOITS THE NETWORK CAPACITY: PERFORMANCE VERSUS REGULARIZATION STRENGTH

In this section, we further illustrate how MIMO better exploits the capacity of the network, through the lens of its sensitivity to regularization.

Our experimental protocol is guided by the following rationale

- Regularization controls the capacity of the network: the higher the regularization, the more constrained the capacity.

- MIMO makes better use of the capacity of the network: the more ensemble members, the more capacity is exploited.

- As a result, MIMO should be more sensitive to the constraining of the capacity of the network. And the more ensemble members (i.e., the larger $M$), the stronger the effect should be.

We consider the ResNet28-10/CIFAR10 and ResNet28-10/CIFAR100 settings used in the main paper where we additionally vary the L1 (respectively L2) regularization while keeping the other L2

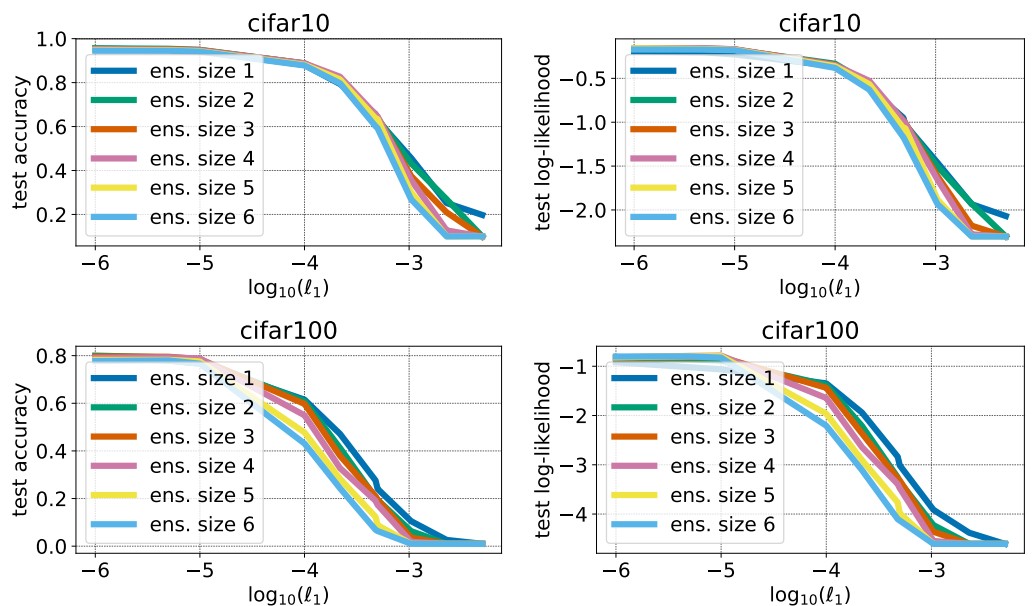

Figure 7: Accuracy and log-likelihood versus varying L1 regularization for ResNet28-10 on CIFAR10 (top row) and CIFAR100 (bottom row). Since MIMO better exploits the capacity of the network, its performance is more sensitive to the constraining of the capacity as the regularization increases. The larger the ensemble size, the stronger the effect.

(respectively L1) term equal to zero. We display in Figures 7-8 the accuracy and log-likelihood over those regularization paths (averaged over three repetitions of the experiments).

As previously hypothesised, we observe that MIMO is indeed more sensitive to the constraining of the capacity of the network as the regularization increases. Moreover, the larger the ensemble size, the stronger the effect. In the case of the L1 regularization, we can show how the accuracy and log-likelihood evolve with respect to the *sparsity* of the network[6]. We report those results in Figure 9 where we can observe the same phenomenon.

## D ADDITIONAL IMAGENET OOD RESULTS

In the following table, we evaluate trained ResNet-50 models on 7 datasets. ImageNet, ImageNet-C, ImageNet-A, and ImageNetV2 each display three metrics: negative log-likelihood, accuracy, and expected calibration error respectively. ImageNet-C further includes mCE (mean corruption error) in parentheses. ImageNet-Vid-Robust, YTTB-Robust, and ObjectNet use their own pre-defined stability metrics.

These experiments were expensive to run, so we were only able to obtain them for a smaller set of methods. We find these results are consistent with the main text's benchmarks, showing MIMO consistently outperforms methods not only on corrupted images, but also across distribution shifts.

| Name | ImageNet | ImageNet-C | ImageNet-A |
|---|---|---|---|
| Deterministic | 0.939 / 76.2% / 0.032 | 3.21 / 40.5% / 0.103 (75.4%) | 8.09 / 0.7% / 0.425 |
| MIMO ($M = 2, \rho = 0.6$) | **0.887 / 77.5% / 0.037** | **3.03 / 43.3% / 0.106 (71.7%)** | **7.76 / 1.4% / 0.432** |
| Wide MIMO ($M = 2, \rho = 0.6$) | 0.843 / 79.3% / 0.061 | 3.1 / 45.0% / 0.150 (69.6%) | 7.52 / 3.3% / 0.46 |

| Name | ImageNetV2 | ImageNet-Vid-Robust | YTTB-Robust | ObjectNet |
|---|---|---|---|---|
| Deterministic | 1.58 / 64.4% / 0.074 | 29.9% | 21.7% | 25.9% |
| MIMO ($M = 2, \rho = 0.6$) | **1.51 / 65.7% / 0.084** | **31.8%** | **22.2%** | **28.1%** |
| Wide MIMO ($M = 2, \rho = 0.6$) | 1.49 / 67.9% / 0.109 | 35.3% | 22.9% | 29.5% |

Table 4: ResNet50/ImageNet & ImageNet OOD: The best single forward pass results are highlighted in bold.

---

[6]A weight is considered to be non zero if its absolute value is larger than $10^{-4}$.

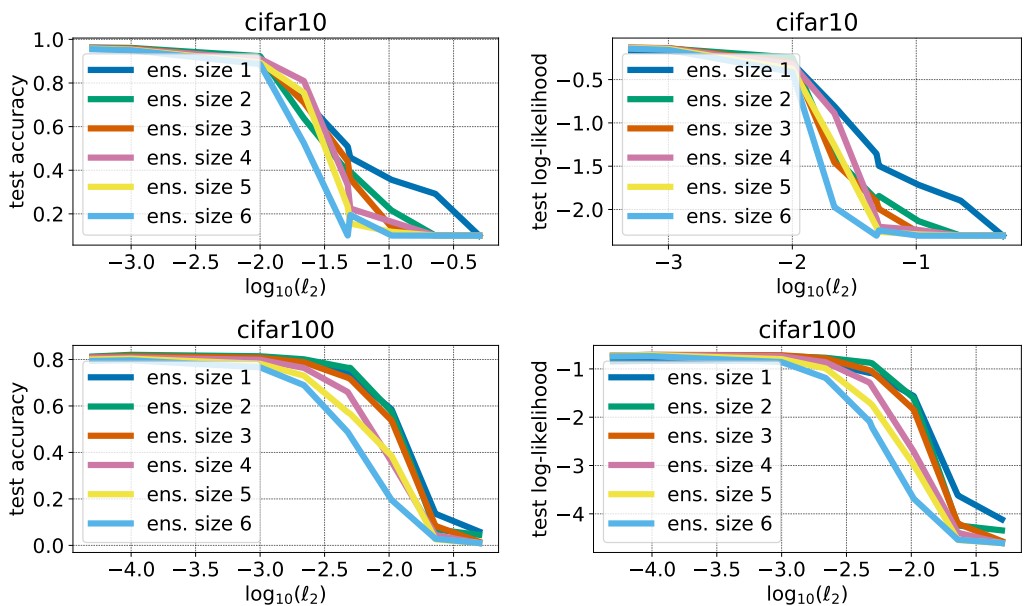

Figure 8: Accuracy and log-likelihood versus varying L2 regularization for ResNet28-10 on CIFAR10 (top row) and CIFAR100 (bottom row). Since MIMO better exploits the capacity of the network, its performance is more sensitive to the constraining of the capacity as the regularization increases. The larger the ensemble size, the stronger the effect.

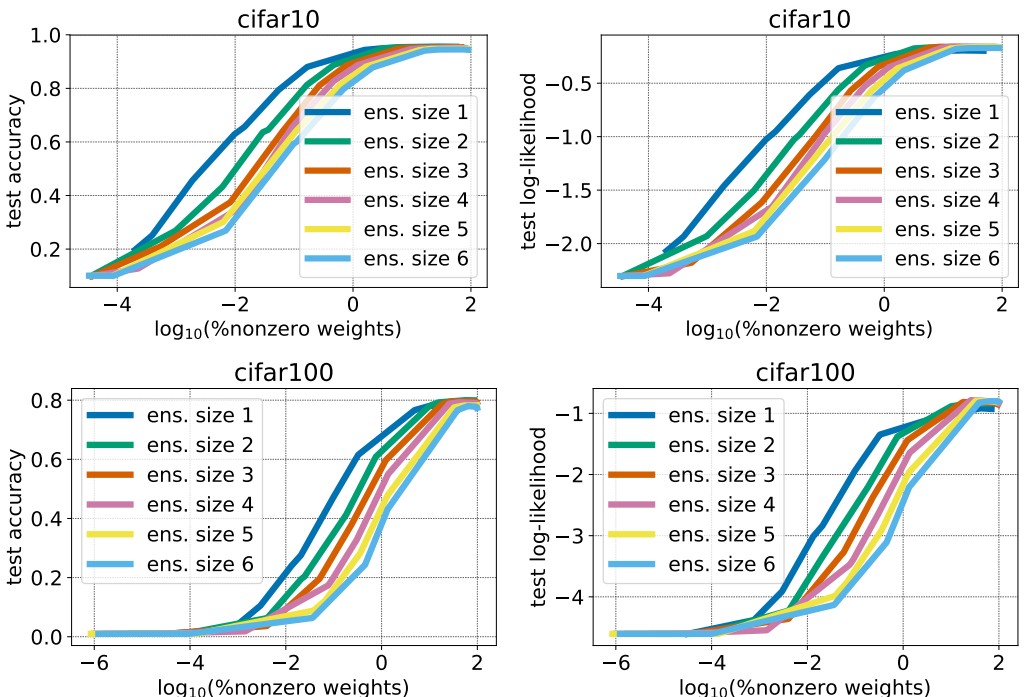

Figure 9: Accuracy and log-likelihood versus varying sparsity of a ResNet28-10 on CIFAR10 (top row) and CIFAR100 (bottom row). Since MIMO better exploits the capacity of the network, its performance is more sensitive to the sparsification of the network (as induced by an increasing L1 regularization). The larger the ensemble size, the stronger the effect.

