# OpenReview forum: "Training independent subnetworks for robust prediction"
_ICLR.cc/2021/Conference — ICLR 2021 Poster_

### Official Review · AnonReviewer3 · 2020-10-22
**Limited novelty**

**Rating:** 6
**Confidence:** 4

**Review:**

Summary: The ensemble method MIMO is proposed in this paper to reduce the inference delay and keep the prediction diversity. Only one model with sufficient capacity exists in this method while multiple implicit subnets are embedded in the parent model. Each subnet has individual I/O, so only one forward pass of the parent model is needed to process all subnets and make the ensemble.

Quality: Medium to high. Pros: 1) The dissections of the subnets are impressive. They design the experiments to survey the loss plane, the parameter/activation projection of different subnets. 2) With the proposed training paradigm and proper test setting, the accuracy improvement can be seen and also uncertainty estimation. 3) They report the SOTA results of accuracy, uncertainty, and robustness on various datasets and their OOD variants when considering the inference latency. Cons: 1) They only report the inference time of one sample, but the total computation costs (e.g. MACs or FLOPS) are omitted. 2) The multiple branch networks are not only used in ensemble and broader usages exist. I know at least three works which involves multi-branch architecture (output-wise or layer-wise) in robustness or knowledge distillation and some of them are also using the ensemble of multiple predictions. The lack of citations in the related work seriously declines the quality of this work.

Clarity: High. Pros: 1) The method framework has a brief and clear explanation (Figure 1). 2) The training methods are conveyed in every detail, including some techniques like “input repetition”. 3) Some critical plots reporting accuracy using error bars or box plots to display the performance variance. Cons: Some format mistakes of symbols are existing. For example, the authors mix the usage of the normal and italic font of “x” when referring to samples in different expressions; the chaotic usage can even happen in the same equation for the first one of Section 3.3.

Originality: Low to medium. Pros: This method successfully uses the multi-branch architecture to reduce the inference delay in ensemble, which is a rather novel idea. Cons: As mentioned above, many similar multi-branch architectures have been used in different but related topics. Consider all of these works, the originality of this work has to be downgraded.

Significance: Low to medium. Pros: The shining points of this work are making ensemble for “free”. They do decrease the inference delay significantly. Cons: 1) The authors attempt to distract our attention on the computational costs of this MIMO architecture and try to make an illusion that it’s convenient in computing. The latency is decreased at the cost of batch size. Other ensemble methods have a longer delay, but they can process a batch of images. This method fills the batch with the same image which implicitly decreases the batch size. Furthermore, no measurement is used on computation costs or other related aspects. I think it’s deceptive and tricky. 2) As shown above, the originality of this work is not as solid as their experiments.

[ Detailed comments]
1. What are the structural considerations for the related work of Section 5 not to be explained in Section 2?
2. In Figure 6, the ‘#’ marked on the abscissa of (b) is redundant

---

> ### Author Response · Authors · 2020-11-17
> **Addressing the concerns**
>
> Thank you for the detailed feedback. We are addressing the two main concerns, originality and computational costs, followed by the minor concerns.
>
> __Main concern - Originality:__ The core contribution of the paper is that by using a multi-input multi-output approach, we can train independent subnetworks within a network and obtain diverse predictions. This addresses the main drawback of previous multi-headed approaches [1,2]: their subnetworks’ predictions highly correlate when using only a single input, as shown in Section 3-4. Since in our setup, independence is ensured by the use of multiple inputs, MIMO also eliminates the need for architectural changes (such as in [1]) which further reduces computational costs. To the best of our knowledge, the idea of using multiple inputs for this purpose is novel, and as we show, it comes with significant computational advantages.
>
> We discuss related work on multi-branch architectures in Section 5. We are keen to expand on this. Could you point us towards the works that we missed?
>
> __Main concern - Computational costs:__ In the paper, we claim that MIMO’s computational cost is equivalent to a standard deep neural network at test time. We support this claim by measuring and reporting the inference delay for all models. In Table 1, 2 and 3 the prediction time column refers to the time it takes to do inference on a single image (ms/example). This metric is calculated over a batch of images---64 for CIFAR10/100 and 128 for ImageNet---and then averaging. The inference delay of MIMO is identical to standard deep neural networks, since they both require a single forward pass for evaluation.
>
> To further address the concern, the number of FLOPS for a ResNet28-10/CIFAR-10 is 10,559M and for MIMO (M=3) it is 10,561M (less than 1% difference). The only additional computational cost is processing the extra inputs and the extra outputs.
>
> Regarding batching, MIMO is fully compatible with batch evaluation. A single input to MIMO has the shape [W, H, MC], where W is the width of the image, H is the height and C is the number of color channels. The third dimension is formed by concatenating the M input images along the channel axis (in the supplied source code, this is done on line 90 in cifar_model.py). At evaluation time, to evaluate MIMO on a batch of 64 images, we first tile the images M times along the channels axis, forming a tensor of shape [64, W, H, MC], execute the 64 forward passes required for MIMO and report the results.
>
> We do not believe that our results are deceptive, let us know in what ways MIMO requires clarification.
>
> __Minor concerns:__
> Notation - we use bold characters to denote random variables and italic to denote their instatiations. Thank you for pointing this out, we are going to clarify this in the paper.
>
> Related works section - we decided to place our related works section after the experiments so that the flow of the paper is uninterrupted. In our opinion, the method is easier to understand this way.
>
> If our reply addressed some of your concerns, please consider updating your score.
>
> [1] Lee, Stefan, et al. "Why M heads are better than one: Training a diverse ensemble of deep networks." arXiv preprint arXiv:1511.06314 (2015).
>
> [2] Tran, Linh, et al. "Hydra: Preserving ensemble diversity for model distillation." arXiv preprint arXiv:2001.04694 (2020).
>
> Edited:
> Updated the number of FLOPS. Our results are in agreement with these results: https://github.com/osmr/imgclsmob/blob/master/chainer_/README.md (WRN-28-10)

---

> ### Author Response · Authors · 2020-11-24
> **Addressing the concerns**
>
> Thank you again R3 for your detailed feedback. We have incorporated them in our revision. If you still have concerns, please let us know. This is the last day for us to reply so we hope we can address any lingering questions.

---

### Official Review · AnonReviewer2 · 2020-10-29
**A clever idea and interesting empirical results**

**Rating:** 6
**Confidence:** 4

**Review:**

This paper proposes to train a single network with M input examples and M corresponding predictions, and the M input examples are mixed to produce the M corresponding predictions. Although only a single network is learned, it implicitly consists of multiple sub-networks due to the nature of multiple inputs and multiple outputs in training. In testing, the single testing example can be replicated M times as inputs, so that M outputs are produced by the trained network. The multiple outputs enable efficient ensembing for robust prediction.

I find the proposed idea very clever. The empirical results on toy data and real data are interesting and compelling. It demonstrates that a single network has the capacity to contain multiple sub-networks, which is an interesting discovery in itself.

I do have two main reservations. The first one is the lacking of some basic, not necessarily rigorous, theoretical formulation and analysis. The second one is about its practical potential. Apparently M has to be rather small, due to the limited capacity of a single network. Its advantage over training multiple networks may not be dramatic.

I am also a bit concerned that the network is trained on M independent examples (although the proposed method does allow for occasional identical examples), but is tested on M identical copies of the same testing example.

I am also unclear about the nature of mixing M independent training examples, and the effect of doing that. That is why I feel some theoretical understanding is needed.

What if we permute the M training examples and check the difference of the corresponding outputs?

---

> ### Author Response · Authors · 2020-11-17
> **Addressing the concerns**
>
> Thank you for the detailed review, we are glad that you found the paper interesting.
>
> Regarding the first reservation, we investigate theoretical questions in the regression case where we show that the advantage of MIMO boils down to a bias-variance tradeoff. This analysis showed that MIMO is particularly effective when the network has excess capacity that MIMO can make use of. We leave further theoretical analysis for future works.
>
> In terms of practical advantage, MIMO significantly improves robustness without increasing the computational costs. We agree that if the test-time computational requirements allow for using a deep ensemble, MIMO might not be the optimal choice, however, when the test-time compute is limited, such as the case in self-driving cars, or large-data applications, MIMO offers considerable advantages over using a standard deep neural network.
>
> Indeed, at test time we repeat the same input to obtain M predictions. This can occur during training (with a small probability), so this is not out-of-distribution for the model. The model behaves as expected, because the subnetworks learn to ignore each other’s inputs during training (since the other inputs provide no useful information for classifying their own input).
>
> We can also permute the inputs. What we see is that they each try to classify their own inputs and ignore the others. Since the subnetworks operate independently, they give different predictions for the same input.

---

### Official Review · AnonReviewer1 · 2020-10-30
**A simple and effective way to train a single network as an ensemble of networks**

**Rating:** 7
**Confidence:** 3

**Review:**

SUMMARY:
This paper describes a multi-input multi-output (MIMO) strategy for training several subnetworks inside a same and single neural network for robust prediction. The approach consists in jointly training the heads to make predictions for their corresponding inputs. The strategy is simple and demonstrates strong results. The experimental study reveals that subnetworks functionally behave as independent networks, hence resulting in a strong and robust ensemble.

STRENGTHS:
- The experimental study is very thorough. I really appreciated the investigation in Section 3, which indeed convincingly shows that the subnetworks behave as independent.
- The method is compared on standard benchmarks, across a wide range of metrics. Experimental results show better performance than single forward pass methods.  Performance is reasonable with respect to a simple solution consisting in training an actual ensemble of 4 networks.
- The paper is well written and easy to follow.

WEAKNESSES:
- I believe the approach to be original, but its similarities/differences with other multi-input multi-output (such as BatchEnsemble) could have been discussed much further to better appreciate the originality of MIMO.
- Although independence between subnetworks is shown empirically, I cannot help but wonder how/why it emerges from the architecture! This is an exciting phenomenon that ought to be better understood. Nevertheless, I believe the actual independence of subnetworks should be nuanced at places (e.g., in the title or in the abstract), as this highly depends on the capacity of architecture and on the problem to solve -- as shown in the experiments themselves.
- Some (hypothetical) theoretical explanations regarding the emergence of independence would have made the paper stronger, although I realize this would be a whole new paper by itself.

---

> ### Author Response · Authors · 2020-11-17
> **Addressing the concerns**
>
> Thank you for the detailed feedback, we are glad that you found our work interesting. We would like to address each weakness in order.
>
> We briefly touched on the relationship between MIMO and BatchEnsemble in our related works section and we are expanding on this a little bit. The reason that MIMO has better diversity is that the subnetworks in MIMO learn to ignore the inputs to the other subnetworks, whereas in BatchEnsemble, it is quite likely that the subnetworks share each other's features due to the high level of parameter sharing.
>
> The key to independence is that the subnetworks do not share features, since features derived from one input contain no useful information for classifying another. As a result, the subnetworks learn a more compact, independent representation for each input, and they learn to classify their corresponding inputs while ignoring the other inputs. We are updating the introduction to bring more attention to this key detail.
>
> Beyond the intuition for independence, we present two forms of empirical analysis to support our claims. First, we look at the loss landscape of the networks and find that the subnetworks converge to different local minima in weight space (Figure 3) and in function space (Figure 4, t-SNE plot). Second, by analysing the conditional variance of the activations within the network, we show that the subnetworks separate within the network and do not share features (Figure 4). We agree that it would be interesting to theoretically analyze the emergence of independence, but leave this for future work.

---

### Official Review · AnonReviewer4 · 2020-11-03
**This paper presents an approach to use a multi-input multi-output configuration for training multiple subnetworks with independent tasks. The authors claim that by ensembling the predictions (output of subnetworks) they can improve model robustness without additional computational cost.**

**Rating:** 8
**Confidence:** 4

**Review:**

Authors assessed how these subnetworks can be as diverse as independently trained networks. The contribution of this paper is in proposing an approach to improve uncertainty estimation and robustness with minor changes (1 percent) to the number of parameters and compute cost.

STRENGTHS:
Training multiple independent subnetworks within a network, with minimal increase in number of parameters.
The use of MIMO makes this approach simple, while it can be evaluated in a single forward pass.

CONCERNS:
The authors claim that the benefits of using multiple predictions can be achieved ‘for free’, while their proposed model increases the number of parameters (even though by 1 percent)
The paper has examined the accuracy and disagreement of the subnetworks, but a detailed evaluation on number of parameters is missing (i.e. where the 1% increase in parameters comes from).
An experiment on more diverse datasets would be also helpful, such as OpenImages.

---

> ### Author Response · Authors · 2020-11-17
> **Addressing the concerns**
>
> Thank you for the review, we are glad that you found our work interesting and impactful.
>
> To address the main concern, the 1% increase in model parameters comes from the extra parameters needed in the first and last layers of MIMO. The first hidden layer requires M times more weights for the M inputs and the output layer also requires M times more weight for the M outputs. Of course, the number of extra weights is architecture dependent, but to give an example, in the case of M=3/ResNet28-10/Cifar-10 they account for 1% additional model parameters. We are clarifying this point in the paper.
>
> We report results on three commonly used image classification benchmarks, Cifar-10, Cifar-100 and ImageNet, with 10, 100 and 1000 classes respectively. It would be interesting to see further datasets and tasks, but we leave this for future works.

---

### Author Response · Authors · 2020-11-24
**Rebuttal Edits**

In response to the reviews, we made a few changes to the paper.
* We clarified where the 1% increase in model parameters come from and also included numbers for FLOPs (Reviewers 3 and 4)
* We clarified the source of independence in the introduction (Reviewer 1)
* We clarified our notation with respect to random variables and their instantiations (Reviewer 4)

---

### Decision · Program_Chairs · 2021-01-07
**Final Decision**

**Decision:**

Accept (Poster)

**Comment:**

This paper proposes a simple but effective method to obtain ensembles of classifiers (almost) for free.
Essentially you train one network on multiple inputs to predict multiple outputs. The authors show that this leads to surprisingly diverse networks - without a significant increase in parameters - which can be used for ensembling during test time.
Because of its simplicity, I can imagine that this approach could become a standard trick in the "deep learning tool chest".

-AC